# Consistency Trajectory Planning: High-Quality and Efficient Trajectory Optimization for Offline Model-Based Reinforcement Learning

**Guanquan Wang**                                      *guanquan-wang@g.ecc.u-tokyo.ac.jp*
*Department of Information and Communication Engineering*
*The University of Tokyo*

**Takuya Hiraoka**                                                  *takuya-h1@nec.com*
*NEC Corporation, Tokyo, Japan*

**Yoshimasa Tsuruoka**                               *yoshimasa-tsuruoka@g.ecc.u-tokyo.ac.jp*
*Department of Information and Communication Engineering*
*The University of Tokyo*

**Reviewed on OpenReview:** *https://openreview.net/forum?id=RVGkT9ISVf*

## Abstract

This paper introduces Consistency Trajectory Planning (CTP), a novel offline model-based reinforcement learning method that leverages the recently proposed Consistency Trajectory Model (CTM) for efficient trajectory optimization. While prior work applying diffusion models to planning has demonstrated strong performance, it often suffers from high computational costs due to iterative sampling procedures. CTP supports few-step trajectory generation without significant degradation in policy quality. We evaluate CTP on the D4RL benchmark and show that it consistently outperforms existing diffusion-based planning methods in long-horizon, goal-conditioned tasks. Notably, CTP achieves higher normalized returns while using fewer denoising steps. In particular, CTP attains comparable—or even superior—performance with reduced inference cost, highlighting its practicality and effectiveness for high-performance, low-latency offline planning.

## 1 Introduction

Recent advances in generative models have significantly impacted offline reinforcement learning (RL), especially in trajectory planning where agents must learn optimal behavior from fixed datasets. Diffusion-based methods such as Diffuser (Janner et al., 2022) and Decision Diffuser (Ajay et al., 2023) have proven effective for modeling complex trajectory distributions, but their reliance on iterative denoising makes real-time decision-making impractical.

To address the inefficiencies inherent in iterative diffusion-based sampling, recent work has explored distilled models such as Consistency Model (CM) (Song et al., 2023), which bypasses the reverse diffusion process by directly learning the mapping from noise to data (Wang et al., 2023; Kang et al., 2024). In the context of model-free RL, CM has demonstrated substantial speedup with only marginal performance degradation (Ding & Jin, 2024; Wang et al., 2024a). However, a critical limitation remains: CM lacks a principled mechanism to trade off between sampling speed and sample quality. This stems from the nature of the distillation process, where the consistency function is trained to project arbitrary intermediate states along the ODE trajectory back to the clean data. As a result, in practice, the multistep sampling procedure of CM for improved sample quality alternates between denoising and injecting noise. This iterative refinement, however, accumulates errors particularly as the number of function evaluations increases.

Building upon this observation, Consistency Trajectory Model (CTM) has recently been proposed as a generalization of both score-based and consistency-based models (Kim et al., 2024). CTM enables anytime-to-anytime transitions along the probability flow ODE, supporting flexible and efficient generation through both short-step and long-jump sampling. CTM retains access to the score function while allowing diverse training losses such as denoising score matching and adversarial losses, ultimately improving expressiveness and generalization. A recent concurrent study (Duan et al., 2025) also investigates the use of CTM for improving inference efficiency in offline RL. The algorithm operates in a *model-free* setting and primarily focuses on policy learning from demonstration data. Although this algorithm accelerates diffusion-based models for decision-making tasks, it offers limited improvements in long-horizon tasks.

Motivated by the strengths of CTM and the need for efficient planning in offline RL, we propose Consistency Trajectory Planning (CTP), a novel offline model-based RL algorithm that integrates CTM into the trajectory optimization process (Section 4). CTP inherits the speed and flexibility of CTM, allowing planners to efficiently navigate the trade-off between planning speed and return quality. Unlike previous score-based planners requiring classifier guidance or iterative refinements, our method enables near single-step sampling (one or a few denoising steps depending on task complexity) while retaining controllability and sample diversity. Furthermore, CTM's access to score information enables conditional planning and return-conditioning without the need for learned Q-functions, which faces challenges due to overestimated Q-values for out-of-distribution actions (Kumar et al., 2020; Levine et al., 2020).

We evaluate CTP on D4RL benchmark tasks (Fu et al., 2020) (Section 5). Across multiple tasks, CTP consistently matches or outperforms prior diffusion-based planners and consistency policies while achieving significant improvements in inference speed—making it well-suited for real-time or high-frequency control applications.

## 2 Related Work

**Diffusion models.** Diffusion models generate data by reversing a progressive noising process (Sohl-Dickstein et al., 2015; Ho et al., 2020). This view unifies with score-based modeling and admits SDE/ODE formulations (Hyvärinen & Dayan, 2005; Song et al., 2021), also connecting to energy-based learning via scores as log-density gradients (Du & Mordatch, 2019; Nijkamp et al., 2019; Grathwohl et al., 2020). Conditioning is commonly implemented via guidance: classifier guidance (Nichol & Dhariwal, 2021) and classifier-free guidance (Ho & Salimans, 2021). These techniques underlie state-of-the-art image/text generation (Saharia et al., 2022; Nichol & Dhariwal, 2021).

**Diffusion Models for Policy Representation.** Diffusion models have been adopted to represent expressive policies in RL (Pearce et al., 2023; Chen et al., 2023). Diffusion-QL (Wang et al., 2023) incorporates diffusion models into both Q-learning and Behavior Cloning to capture multi-modal action distributions, with later work improving efficiency via action approximation to avoid repeated denoising during training (Kang et al., 2024). Other work includes robotic visuomotor diffusion policies with receding-horizon control and visual conditioning (Chi et al., 2023); IDQL, which reinterprets IQL as actor–critic and extracts policies from diffusion-parameterized behavior models (Hansen-Estruch et al., 2023); QSM, which links diffusion-policy scores to Q-gradients for principled updates (Psenka et al., 2024); DQS, which samples from Boltzmann policies via diffusion (Jain et al., 2025); and DACER, which applies diffusion policies to online actor–critic with GMM-based entropy estimation (Wang et al., 2024b).

**Diffusion Models for Trajectory Optimization.** In addition to representing policies, diffusion models have also been employed as trajectory generators for planning in offline RL (Liang et al., 2023; Du et al., 2023; Yang et al., 2023; Li et al., 2023; Chen et al., 2024; Dong et al., 2024). A notable example is Diffuser (Janner et al., 2022), which learns to generate entire trajectories from offline data and applies guidance to bias the trajectories toward high returns or task-specific goals. At execution time, however, only the first action of each generated trajectory is applied, after which the model replans in a receding-horizon fashion. Decision Diffuser (Ajay et al., 2023) follows a similar idea but conditions trajectory generation directly on target returns or goals, eliminating the need for an auxiliary reward model.

**Summary.** Across behavior cloning, actor–critic, energy-based policies, and online control (Chi et al., 2023; Hansen-Estruch et al., 2023; Psenka et al., 2024; Jain et al., 2025; Wang et al., 2024b), diffusion-based methods typically require multi-step denoising or gradient refinements, incurring nontrivial inference latency; trajectory planners like Diffuser/Decision Diffuser further amplify cost by generating full sequences and replanning. While Diffusion-QL variants reduce some training-time overhead, they do not fundamentally eliminate the need for iterative reverse diffusion processes during the inference process.

**Positioning of our work.** Beyond diffusion-based planners, the work most closely related to ours is Consistency Planning (CP) (Wang et al., 2024a), which first introduced the use of consistency models for trajectory optimization in offline RL. Both CP and our proposed CTP share the same high-level goal of trajectory-level planning; however, CTP introduces several key differences. First, instead of using a standard CM that only learns mappings from $x_t$ to $x_0$, CTP leverages CTM which support anytime-to-anytime mappings along the probability flow ODE. This formulation enables greater flexibility at inference time and reduces reliance on iterative guidance. Second, whereas CP steers trajectory generation using classifier-free guidance, CTP adopts a critic-based selection mechanism that evaluates multiple candidate plans and selects the best one according to predicted returns (Chi et al., 2023). This avoids the potential instability of classifier-based guidance while aligning the planner directly with the RL objective. In addition, CTP incorporates several architectural refinements, including stride-based trajectory representation and transformer backbones, which further improve scalability to complex benchmarks. Overall, these modifications allow CTP to handle more challenging tasks while maintaining the efficiency advantages of the consistency framework.

Motivated by these limitations, our work introduces a more efficient alternative based on the recently proposed CTM, which supports one or a few denoising steps, depending on task complexity (single-step on Maze2D, and two steps for harder benchmarks). By integrating CTM into the trajectory optimization process, our method circumvents the need for iterative denoising and enables fast planning with minimal performance degradation. In contrast to prior diffusion-based planners, CTP achieves a favorable trade-off between sample quality and computational cost, making it particularly well-suited for time-sensitive offline RL applications.

## 3 Preliminary

### 3.1 Reinforcement Learning Problem Setting

We consider standard reinforcement learning formulated as a Markov Decision Process (MDP) $M = (S, A, P, R, \gamma, d_0)$, where $S$ is the state space, $A$ is the action space, $P$ is the transition function, $R$ is the reward, $\gamma \in [0, 1)$ is the discount factor, and $d_0$ is the initial state distribution. The objective is to learn a policy that maximizes the expected discounted return $\mathbb{E}\left[\sum_{k=0}^{k_{end}} \gamma^k r(s_k, a_k)\right]$, where $k_{end}$ is the index of final time step in a trajectory.

### 3.2 Consistency Trajectory Models

Diffusion models generate samples by gradually adding and then removing Gaussian noise, but require hundreds of denoising steps at inference, limiting their practicality in RL. CMs (Song et al., 2021; Kim et al., 2024) address this by learning a direct mapping from noisy input $\mathbf{x}_t$ at arbitrary noise level $t$ back to its clean counterpart $\mathbf{x}_\epsilon$, enabling generation in one or a few steps. This efficiency makes CMs appealing for planning tasks where inference speed is critical.

CTMs extend the idea of CMs (Song et al., 2021; Kim et al., 2024) from data generation to trajectory prediction. The framework follows a teacher–student distillation paradigm. The teacher model $D_\phi$ is trained as a denoiser, mapping noisy data back to clean ones:

$$\mathcal{L}(\phi) := \mathbb{E}_{\sigma \sim p_{train}, \tau \sim \mathcal{D}, n \sim \mathcal{N}(0,\sigma^2 \mathrm{I})} \left[\|D_\phi(\mathbf{x}_\sigma(\tau), \sigma) - \mathbf{x}_0(\tau)\|_2^2\right], \tag{1}$$

where the noise level $\sigma$ is sampled from a log-normal distribution $p_{\text{train}}$ following Karras et al. (2022).

To stabilize training, $D_{\boldsymbol{\phi}}$ adopts a skip-connection parameterization:

$$D_{\boldsymbol{\phi}}(\mathbf{x}_t, t) = c_{\text{skip}}(t)\,\mathbf{x}_t + c_{\text{out}}(t)\,F_{\boldsymbol{\phi}}(\mathbf{x}_t, t), \tag{2}$$

where $F_{\boldsymbol{\phi}}$ is a neural network and $c_{\text{skip}}$, $c_{\text{out}}$ are time-dependent coefficients that ensure correct boundary behavior at $t = \epsilon$. These coefficients are chosen such that $c_{\text{skip}}(\epsilon) = 1$ and $c_{\text{out}}(\epsilon) = 0$, ensuring that the model exactly reproduces the clean data at the boundary $t = \epsilon$, i.e., $D_{\boldsymbol{\phi}}(\mathbf{x}_\epsilon, \epsilon) \equiv \mathbf{x}_0$.

The student model $G_{\boldsymbol{\theta}}$ (Eq. 3) learns to map noisy inputs at arbitrary $t$ to less noisy versions at $w < t$, combining an identity shortcut with a neural residual.

$$G_{\boldsymbol{\theta}}\left(\mathbf{x}_t, t, w\right) = \frac{w}{t}\mathbf{x}_t + \left(1 - \frac{w}{t}\right) g_{\boldsymbol{\theta}}\left(\mathbf{x}_t, t, w\right), \tag{3}$$

Distillation enforces local and global consistency by comparing student predictions with targets obtained from the teacher through short ODE integrations, typically solved with a Heun method. This yields the CTM loss (Eq. 4), while denoising score matching (Eq. 5) regularizes cases where $w \approx t$.

$$\mathcal{L}_{\text{CTM}}(\boldsymbol{\theta}; \boldsymbol{\phi}) := \mathbb{E}_{t \in [0, t_N]} \mathbb{E}_{w \in [0, t]} \mathbb{E}_{u \in [w, t)} \mathbb{E}_{\mathbf{x}_0} \mathbb{E}_{\mathbf{x}_t | \mathbf{x}_0} \left[ d\left(\mathbf{x}_{\text{target}}\left(\mathbf{x}_t, t, u, w\right), \mathbf{x}_{\text{est}}\left(\mathbf{x}_t, t, w\right)\right) \right], \tag{4}$$

where $d(\cdot, \cdot)$ is the squared $\ell_2$ distance with $d(x, y) = \|(x - y)\|_2^2$, and $\mathbf{x}_{\text{est}}\left(\mathbf{x}_t, t, w\right) := G_{\text{sg}(\boldsymbol{\theta})}\left(G_{\boldsymbol{\theta}}\left(\mathbf{x}_t, t, w\right), w, 0\right)$, $\mathbf{x}_{\text{target}}\left(\mathbf{x}_t, t, u, w\right) := G_{\text{sg}(\boldsymbol{\theta})}\left(G_{target}, w, 0\right)$, representing the predictions at time $0$.

$$\mathcal{L}_{\text{DSM}}(\boldsymbol{\theta}) = \mathbb{E}_{t, \mathbf{x}_0} \mathbb{E}_{\mathbf{x}_t | \mathbf{x}_0} \left[ \|\mathbf{x}_0 - g_{\boldsymbol{\theta}}\left(\mathbf{x}_t, t, t\right)\|_2^2 \right], \tag{5}$$

which acts as a regularization mechanism that strengthens the learning signal and improve the accuracy when $w \approx t$.

To further improve sample quality, an adversarial discriminator is added, yielding a GAN loss (Eq. 6).

$$\mathcal{L}_{\text{GAN}}(\boldsymbol{\theta}, \boldsymbol{\eta}) = \mathbb{E}_{\mathbf{x}_0} \left[ \log d_{\boldsymbol{\eta}}\left(\mathbf{x}_0\right) \right] + \mathbb{E}_{t \in [0, t_N]} \mathbb{E}_{w \in [0, t]} \mathbb{E}_{\mathbf{x}_0} \mathbb{E}_{\mathbf{x}_t | \mathbf{x}_0} \left[ \log\left(1 - d_{\boldsymbol{\eta}}\left(\mathbf{x}_{\text{est}}\left(\mathbf{x}_t, t, w\right)\right)\right) \right], \tag{6}$$

where $d_{\boldsymbol{\eta}}$ is the discriminator function, and $\boldsymbol{\eta}$ denotes its parameters.

The overall training objective combines these components (Eq. 7), and the complete training algorithm is summarized in Appendix C.

$$\mathcal{L}(\boldsymbol{\theta}, \boldsymbol{\eta}) := \mathcal{L}_{\text{CTM}}(\boldsymbol{\theta}; \boldsymbol{\phi}) + \lambda_{\text{DSM}} \mathcal{L}_{\text{DSM}}(\boldsymbol{\theta}) + \lambda_{\text{GAN}} \mathcal{L}_{\text{GAN}}(\boldsymbol{\theta}, \boldsymbol{\eta}). \tag{7}$$

## 4 Planning with Consistency Trajectory Model

This paper explores the integration of CTM into the planning architecture in offline RL. In the following, we discuss how we use CTM for the trajectory optimization process. Section 4.1 details the training process of each component, and Section 4.2 describes how the consistency trajectory planner is applied during inference.

### 4.1 Training process

**Trajectory representation.** As outlined by Ajay et al. (2023), the diffusion process encompasses only the state transitions, as described by

$$\mathbf{x}_{t_i}(\tau) := (s_k, s_{k+M}, \ldots, s_{k+(H-1)M})_{t_i}. \tag{8}$$

In this notation, $k$ indicates the timestep of a state within a trajectory $\tau$, $H$ represents the planning horizon, and $t_i \in [\epsilon, t_N]$ is the timestep in the diffusion sequence. To make the model look ahead farther, we choose a jump-step planning strategy (Lu et al., 2025). Jump-step planning models $H \times M$ environment steps, where $M \in N_+$ is the planning stride. Consequently, $\mathbf{x}_{t_i}(\tau)$ is defined as a noisy sequence of states, represented as a two-dimensional array where each column corresponds to a different timestep of the trajectory. In the training process, the sub-sequence $t_i$ follows the Karras boundary schedule (Karras et al., 2022):

$$t_i = \left( \epsilon^{1/\rho} + \frac{i-1}{N-1} \left( t_N^{1/\rho} - \epsilon^{1/\rho} \right) \right)^{\rho}, \tag{9}$$

where $\epsilon = 0.002$, $t_N = 80$, and $\rho = 7$.

**Inverse dynamics model training.** To derive actions from the states generated by the diffusion model, we employ an inverse dynamics model (Agrawal et al., 2016; Pathak et al., 2018), denoted as $h_{\boldsymbol{\varphi}}$, trained using $(s_k, s_{k+M})$ as input, $a_k$ as target, sampled from the dataset $\mathcal{D}$ consisting of trajectories. Therefore, actions can be obtained via the inverse dynamics model by extracting the state tuple $(s_k, s_{k+M})$ at diffusion timestep $t_0$:

$$\mathcal{L}(\boldsymbol{\varphi}) := \mathbb{E}_{(s_k, a_k, s_{k+M}) \sim \mathcal{D}} \left[ \| a_k - h_{\boldsymbol{\varphi}}(s_k, s_{k+M}) \|_2^2 \right]. \tag{10}$$

**CTM training.** Following the general framework in Section 3.2, we instantiate CTM in RL using $\mathbf{x}_{t_i}(\tau)$ as noisy sequences (Eq. 8). The teacher $D_{\boldsymbol{\phi}}$ is trained to map these noisy trajectories back to clean ones using the objective in Eq. 1, and knowledge is distilled to the student $G_{\boldsymbol{\theta}}$ via the CTM and auxiliary losses (Eqs. 4–7). This adaptation allows CTM to directly generate candidate state sequences conditioned on the current state $s_k$, which are later ranked by the critic model.

**Critic Model training.** During the sampling process, we use Monte Carlo sampling from selections, where $C$ selections are firstly sampled from CTM as candidates. To select the best plan from the $C$ selections, a critic model $V_{\boldsymbol{\alpha}}$ is trained using $\mathbf{x}_{t_0}(\tau)$ as input, the accumulated discounted returns $R_k$ as target output, where $\boldsymbol{\alpha}$ denotes the parameters of the critic network. Specifically, $R_k$ is calculated by

$$R_k = \sum_{h=0}^{t_{end}} \gamma^h r_{k+h}. \tag{11}$$

To train the critic model $V_{\boldsymbol{\alpha}}$, we minimize the mean squared error between the predicted return and the actual accumulated return:

$$\mathcal{L}_{\text{critic}}(\boldsymbol{\alpha}) = \mathbb{E}_{\tau \sim \mathcal{D}} \left[ \left( V_{\boldsymbol{\alpha}}(\mathbf{x}_{t_0}(\tau)) - R_k \right)^2 \right]. \tag{12}$$

This value-based selection process enables the planner to rank candidate trajectories according to their expected returns and choose the highest-scoring plan for execution.

### 4.2 Inference process

**Sampling with CTM.** To generate samples using CTM, we follow a reverse-time denoising procedure along a predefined sequence of time steps $t_0, t_1, \ldots, t_N$, where $t_n = \frac{n}{N} t_N$ and $t_0 = \epsilon$. Sampling begins by drawing an initial noisy sample $\mathbf{x}_{t_N}$ from a standard Gaussian prior $\mathcal{N}(\mathbf{0}, t_N^2 \mathbf{I})$. Then, for each time step

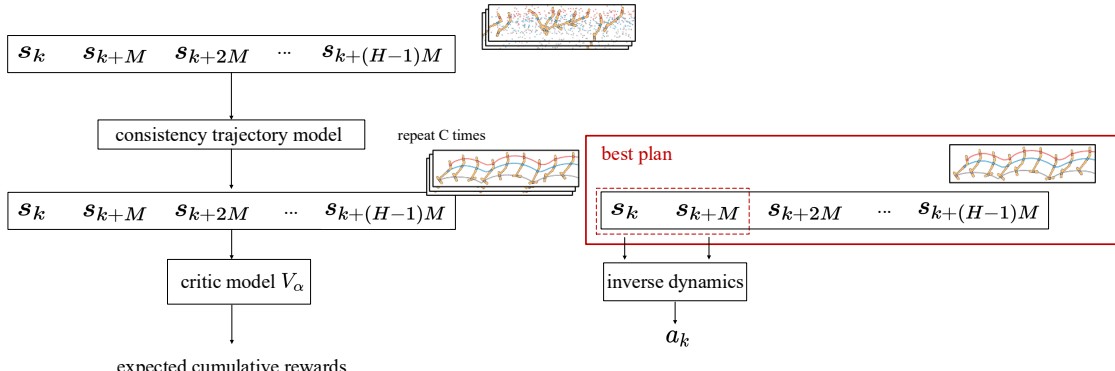

Figure 1: Consistency Trajectory Planning. Given the current state $s_k$, Consistency Trajectory Planning generates $C$ sequences of future states with planning horizon $H$. Then, the best plan is selected by the critic model. Finally, the inverse dynamics model is used to extract and execute the action $a_k$ from $s_k$ and $s_{k+M}$ which are from the selected best plan.

$n = N - 1, \ldots, 0$, the model applies a single-step denoising operation using the student model $G_{\boldsymbol{\theta}}$, which maps the current noisy sample $\mathbf{x}_{t_{n+1}}$ from time $t_{n+1}$ to a less noisy representation at time $t_n$.

Formally, this process iteratively computes

$$\mathbf{x}_{t_n} \leftarrow G_{\boldsymbol{\theta}}(\mathbf{x}_{t_{n+1}}, t_{n+1}, t_n), \tag{13}$$

until reaching the final output $\mathbf{x}_{t_0}$, which serves as the generated sample. Unlike the inference process in CM, we do not add noise perturbations to intermediate samples, since such perturbations would cause error accumulation. This deterministic sampling procedure allows CTM to produce high-quality samples in a small number of steps, while preserving flexibility in the choice of discretization schedule.

---

**Algorithm 1** Consistency Trajectory Planning

1: **Input**: Planning horizon $H$, Dataset $\mathcal{D}$, Discount factor $\gamma$, Candidate number $C$, Planning stride $M$
2: **Initialize**: Diffusion Transformer Planner $D_{\boldsymbol{\phi}}$, Consistency Trajectory Planner $g_{\boldsymbol{\theta}}$, Diffusion Inverse dynamics $h_{\boldsymbol{\varphi}}$, Critic $V_{\boldsymbol{\alpha}}$
3: Calculate accumulated discounted returns $R_k = \sum_{h=0}^{\text{end}} \gamma^h r_{k+h}$ for every step $k$
4: **function** TRAINING
5:     Sample $s_k, s_{k+M}, \ldots, s_{k+(H-1)M}, a_k, a_{k+M}, \ldots, a_{k+(H-1)M}, R_k$ from $\mathcal{D}$
6:     Train diffusion model $D_{\boldsymbol{\phi}}$ using $s_k$ as condition and $s_{k,k+M,\ldots,k+(H-1)M}$ as target output (Eq. 1)
7:     Distill consistency trajectory planner $g_{\boldsymbol{\theta}}$ (Eq. 7)
8:     Train inverse dynamics $h_{\boldsymbol{\varphi}}$ using $s_k, s_{k+M}$ as input, $a_k$ as target output (Eq. 10)
9:     Train critic $V_{\boldsymbol{\alpha}}$ using $s_{k,k+M,\ldots,k+(H-1)M}$ as input, $R_k$ as target output (Eq. 12)
10: **end function**
11: **function** PLANNING($s$)
12:     Randomly generate $C$ plans using CTP sampling, while fixing the first state as $s$ during sampling
13:     Select the best plan using critic $V_{\boldsymbol{\alpha}}$
14:     Use the inverse dynamics $h_{\boldsymbol{\varphi}}$ to generate action using $s$ and the next state in the best plan
15: **end function**

---

**CTP Inference.** During the inference process, we first observe a state $s$ in the environment and sample a Gaussian noise $\mathbf{x}_{t_N}$. Then, CTM iteratively predicts the denoised trajectories $\mathbf{x}_{t_n}$ from the noisy inputs. This process is repeated $C$ times to generate $C$ plans using $g_{\boldsymbol{\theta}}$, while the first state $s$ is fixed during sampling. Then, we select the best plan using critic $V_{\boldsymbol{\alpha}}$. Finally, we extract states $(s_k, s_{k+M})$ from the denoised

trajectory and get the action $a_k$ via our inverse dynamics model $h_\phi$. The algorithm of CTP is provided in Algorithm 1 and visualized in Figure 1.

## 5   Experiment

In this section, we present the experiment environment, experiment setting and report empirical results that validate the effectiveness of the proposed CTP algorithm across diverse offline RL tasks.

### 5.1   Experiment Environment

We train the diffusion model, inverse dynamics model, and consistency trajectory model on publicly available D4RL datasets. Evaluation is conducted across diverse Gym tasks, including locomotion (HalfCheetah, Hopper, Walker2d), long-horizon planning (Maze2D), and high-dimensional robotic control tasks (Antmaze, Kitchen, Adroit) from the D4RL benchmark suite (Fu et al., 2020). These tasks are characterized by continuous state and action spaces and are conducted under offline RL settings. More details of the tasks are provided in Appendix A.

**Locomotion** The locomotion tasks—Hopper, HalfCheetah, and Walker2d—are widely adopted due to their controlled dynamics, reproducibility, and varying levels of complexity. These environments are based on the MuJoCo physics engine (Todorov et al., 2012) and simulate planar bipedal or quadrupedal agents that must learn to move forward efficiently.

**Maze2D** To validate the long-horizon planning capabilities of CTP, we conduct an evaluation in the Maze2D environment (Fu et al., 2020), where the task involves navigating to a specific goal location, with a reward of 1 assigned only upon reaching the goal. Because it requires hundreds of steps to reach the goal, even the most advanced model-free algorithms struggle with effective credit assignment and consistently reaching the goal.

**AntMaze** The AntMaze task extends the Maze2D environment by replacing the simple 2D ball agent with a more complex quadrupedal "Ant" robot, thereby combining challenges of both locomotion and high-level planning. In the diverse variant of the dataset, the ant is initialized at random positions and directed toward randomly sampled goals. In contrast, the play variant consists of trajectories guided toward manually selected goal locations within the maze.

**Kitchen** The Kitchen environment simulates a robotic manipulator interacting with various appliances in a realistic kitchen setting, requiring the agent to perform multi-stage manipulation tasks to accomplish specified goals. In the partial dataset, only a subset of trajectories successfully achieves the full task, allowing imitation-based methods to benefit from selectively identifying informative demonstrations. The mixed dataset, on the other hand, contains no trajectories that solve the task in its entirety, necessitating the use of RL to stitch together relevant sub-trajectories.

**Adroit** The Adroit Hand benchmark consists of four high–degree-of-freedom manipulation tasks constructed from motion-capture demonstrations of human hand movements (Rajeswaran et al., 2018; Fu et al., 2020). The benchmark couples challenging planning objectives with low-level motor control, thereby providing a stringent test of our approach.

### 5.2   Experiment Setting

We evaluate the performance of our proposed method by comparing it against both actor-critic-based approaches, including Consistency Actor-Critic (C-AC) (Ding & Jin, 2024) and Diffusion-QL (D-QL) (Wang et al., 2023), as well as model-based planning methods, such as Diffuser (Janner et al., 2022), Decision Diffuser (DD) (Ajay et al., 2023), Consistency Planning (CP) (Wang et al., 2024a), Reward-Aware Consistency Trajectory Distillation (RACTD) (Duan et al., 2025) and Lower Expectile Q-learning (LEQ) (Park & Lee, 2025).

Table 1: The average scores of Diffuser, Decision Diffuser, Diffusion-QL, Consistency-AC, Consistency Planning and our method on D4RL locomotion tasks are shown. The results of previous work are quoted from Ding & Jin (2024), Ajay et al. (2023) and Wang et al. (2024a).

| Dataset | Environment | Diffuser | DD | D-QL | C-AC | CP | CTP |
|---------|-------------|----------|------|------|------|------|------|
| Medium-Expert | Halfcheetah | 79.8 | 90.6 | **96.8** | 84.3 | 94 | $89.3 \pm 0.6$ |
| Medium-Replay | Halfcheetah | 42.2 | 39.3 | 47.8 | **58.7** | 40.6 | $43.4 \pm 0.4$ |
| Medium | Halfcheetah | 44.2 | 49.1 | 51.1 | **69.1** | 46.8 | $50.4 \pm 0.1$ |
| Medium-Expert | Hopper | 107.2 | **111.8** | 111.1 | 100.4 | 107.5 | $107.5 \pm 1.2$ |
| Medium-Replay | Hopper | 96.8 | 100 | **101.3** | 99.7 | 97.8 | $90.0 \pm 1.0$ |
| Medium | Hopper | 58.5 | 79.3 | **90.5** | 80.7 | 87.8 | $83.6 \pm 1.3$ |
| Medium-Expert | Walker2d | 108.4 | 108.8 | 110.1 | **110.4** | 109.8 | $110.1 \pm 0.05$ |
| Medium-Replay | Walker2d | 61.2 | 75 | **95.5** | 79.5 | 75.3 | $86.9 \pm 0.3$ |
| Medium | Walker2d | 79.7 | 82.5 | **87.0** | 83.1 | 80.5 | $85.7 \pm 0.2$ |
| **Average** | - | 75.3 | 81.8 | **87.9** | 85.1 | 82.2 | 83 |

Table 2: The performance of CTP, Diffuser, and previous model-free algorithms in the Maze2D environment, which tests long-horizon planning due to its sparse reward structure. The results of previous work are quoted from the data provided in Janner et al. (2022), Wang et al. (2024a), Lu et al. (2025) and Duan et al. (2025).

| Dataset | MPPI | CQL | IQL | Diffuser | CP | RACTD | D-QL | CTP |
|---------|------|-----|-----|----------|------|-------|------|------|
| Maze2D U-Maze | 33.2 | 5.7 | 47.4 | 113.9 | 122.7 | 125.7 | 140.6 | **154.1** $\pm 2.3$ |
| Maze2D Medium | 10.2 | 5.0 | 34.9 | 121.5 | 121.4 | 130.8 | 152.0 | **167.1** $\pm 2.4$ |
| Maze2D Large | 5.1 | 12.5 | 58.6 | 123.0 | 119.5 | 143.8 | 186.8 | **216.7** $\pm 3.4$ |
| Average | 16.2 | 7.7 | 47.0 | 119.5 | 121.2 | 133.4 | 159.8 | **179.3** |

For all experiments, we report results as the mean over 150 independent planning seeds to ensure statistical robustness. Following the evaluation protocol established in Fu et al. (2020), we adopt the normalized average return as the primary performance metric.

Unless otherwise specified, the CTP algorithm employs $N = 2$ denoising steps, which we found to yield near-saturated performance across most tasks. An exception is the Maze2D domain and Walker2d, where we use a single denoising step ($N = 1$) due to the relative simplicity of the environment. We use $N = 3$ steps for the Kitchen domain. For comparison, competing methods employ different denoising schedules: the diffusion policy utilizes $N = 5$ steps (Wang et al., 2023), Diffuser adopts $N = 20$ steps, and Decision Diffuser uses $N = 40$ steps (Janner et al., 2022; Ajay et al., 2023). Consistency-based baselines such as Consistency Actor-Critic and Consistency Planning are evaluated with $N = 2$ denoising steps, consistent with their original implementations. More implementation details are provided in Appendix C.

### 5.3 Experimental Results

As shown in Table 1, CTP achieves competitive performance across standard locomotion benchmarks. These results highlight the effectiveness of CTP in matching the performance of prior state-of-the-art diffusion-based planning algorithms.

Given the relatively lower complexity of Maze2D compared to AntMaze and Kitchen, our model is capable of generating effective trajectories with one-step generation ($N = 1$), while still achieving best performance (Table 2). Notably, when compared to the recent concurrent method RACTD (Duan et al., 2025) and D-

Table 3: The performance of CTP, Diffuser, and previous model-free algorithms in the Kitchen environment, which tests both locomotion and high-level planning capability. The results of previous work are derived from the data provided in Ding & Jin (2024) and Ajay et al. (2023).

| Dataset | Environment | Diffuser | DD | D-QL | C-AC | CTP |
|---------|-------------|----------|-----|------|------|-----|
| Mixed | Kitchen | 47.5 | $65 \pm 2.8$ | $62.6 \pm 5.1$ | $45.8 \pm 1.5$ | $\mathbf{74.5} \pm 0.3$ |
| Partial | Kitchen | 33.8 | $57 \pm 2.5$ | $60.5 \pm 6.9$ | $38.2 \pm 1.8$ | $\mathbf{91.2} \pm 1.0$ |
| **Average** | - | 40.65 | 61 | 61.55 | 42 | **82.85** |

Table 4: The performance of CTP, D-QL and LEQ in the Antmaze environment. The results of D-QL and LEQ are quoted from (Wang et al., 2023) and Park & Lee (2025).

| Dataset | Environment | LEQ | D-QL | CTP |
|---------|-------------|-----|------|-----|
| Diverse | Antmaze-Large | $60.2 \pm 18.3$ | $56.6 \pm 7.6$ | $\mathbf{82.0} \pm 3.1$ |
| Play | Antmaze-Large | $62.0 \pm 9.9$ | $46.4 \pm 8.3$ | $\mathbf{82.0} \pm 3.1$ |
| Diverse | Antmaze-Medium | $46.2 \pm 23.2$ | $78.6 \pm 10.3$ | $\mathbf{86.0} \pm 2.8$ |
| Play | Antmaze-Medium | $76.3 \pm 17.2$ | $76.6 \pm 10.8$ | $\mathbf{83.3} \pm 3.0$ |
| **Average** | - | 61.18 | 64.6 | **83.33** |

QL (Wang et al., 2023), which adopt model-free approach, CTP consistently achieves higher returns across all Maze2D tasks, which require accurate trajectory optimization and modeling of long-term dependencies. By leveraging environment dynamics through CTM, CTP enables high-quality planning under sparse-reward, long-horizon scenarios.

Tables 3 and 4 present the performance of CTP on the Kitchen and AntMaze benchmarks, illustrating its robustness in addressing complex, goal-conditioned control tasks. Our experimental results demonstrate that CTP performs competitively not only in relatively simple environments such as Maze2D—where no robotic actuation is involved—but also in more challenging domains like Kitchen and AntMaze, which demand high-dimensional, temporally coordinated control.

We further evaluate CTP on the "expert" dataset from the Adroit benchmark, which consists of 5000 trajectories generated by a policy that consistently completes the task. Quantitative results are reported in Table 5. While CTP does not outperform all prior methods across every task, it achieves competitive results overall. These findings suggest that CTP is effective in goal-conditioned manipulation scenarios that require precise spatial reasoning and long-horizon planning.

While CTP achieves competitive overall results, we note that model-free Diffusion-QL obtains stronger scores on several MuJoCo locomotion tasks (Table 1). This performance gap can be explained by the nature of these benchmarks. First, locomotion tasks such as HalfCheetah, Hopper, and Walker2d primarily reward short-horizon control signals (e.g., maintaining balance and maximizing forward velocity), which can be effectively optimized through direct policy gradient updates and critic-based temporal-difference learning (Wang et al., 2023; Ding & Jin, 2024). In contrast, model-based planning approaches like CTP introduce additional sources of approximation error due to dynamics modeling and trajectory sampling, which may be unnecessary overhead in these relatively simple tasks. Second, trajectory-level planning with small candidate counts $C$ may under-explore the fine-grained action variations that are critical for maximizing continuous control rewards in MuJoCo environments. Third, the critic in CTP is trained to evaluate full trajectories rather than single-step actions, which can be less sample-efficient for dense-reward settings compared to

Table 5: The performance of CTP against other baselines in the Adroit environment. The results of previous work are quoted from the data provided in He et al. (2024) and Lu et al. (2025).

| Dataset | Environment | BC | BCQ | CQL | IQL | AlignIQL | D-QL | CTP |
|---------|-------------|------|-------|-------|-------|----------|-------|-----------------|
| Expert | door | 34.9 | 99.0 | 101.5 | 103.8 | **104.6** | 104.3 | **104.1** $\pm 0.5$ |
| Expert | hammer | **125.6** | 107.2 | 86.7 | 116.3 | **124.7** | 55.9 | $110.5 \pm 3.1$ |
| Expert | pen | 85.1 | 114.9 | 107.0 | 111.7 | **116.0** | 60.9 | $104.1 \pm 5.3$ |
| Expert | relocate | 101.3 | 41.6 | 95.0 | 102.7 | 106.0 | 108.8 | **108.8** $\pm 0.6$ |
| **Average** | - | 86.7 | 90.7 | 97.6 | 108.6 | **112.8** | 82.48 | 106.88 |

Figure 2: Comparison of the average normalized scores (top row) and inference times (bottom row) of CTP and Diffuser on three walker2d datasets: walker2d-medium-expert, walker2d-medium, and walker2d-medium-replay. Each column corresponds to a specific dataset. The x-axis denotes the number of denoising steps $N$ on a logarithmic scale. Vertical error bars represent the standard deviation over five random seeds. While CTP achieves near-optimal performance with significantly fewer denoising steps (e.g., saturating at $N = 1$), Diffuser requires substantially more steps to match similar scores. Inference time grows rapidly with larger $N$ for Diffuser, whereas CTP maintains consistently low inference time.

actor-critic methods tailored for stepwise feedback. Together, these factors suggest that the gap arises not from a fundamental weakness of CTP, but from a mismatch between algorithmic design and the structural simplicity of MuJoCo tasks.

**Computational Time.** To assess the computational efficiency of CTP relative to Diffuser under varying denoising step numbers $N$, we conduct a systematic evaluation of both inference time and policy performance on the walker2d-medium-exper task. Since both CTP and Diffuser are built upon generative modeling frameworks based on probability flow, their inference cost scales with the number of denoising steps $N$. However, by design, the CTP enables effective few-step sampling, in contrast to diffusion models which typically require iterative refinement to achieve comparable sample quality. Figure 2 presents a detailed comparison of the average normalized scores and inference time (in milliseconds per sample) across varying

denoising steps. Each data point reports the mean and standard deviation computed over five random seeds. As shown, CTP achieves near-optimal performance with $N = 1$, and saturates fully by $N = 2$. In contrast, Diffuser requires up to $N = 20$ steps to reach its performance plateau, with inference time increasing significantly with each additional step. Furthermore, Diffuser with $N = 20$ requires approximately 1900 ms per sample for inference, while CTP with $N = 1$ achieves the same result within approximately 15 ms, demonstrating its substantial efficiency under this setting. Despite this large discrepancy in computation time, CTP continues to outperform Diffuser in terms of policy quality, yielding a slight improvement in normalized score. To further validate scalability, we also evaluate inference latency on large-scale tasks beyond locomotion. Figure 3 reports results on AntMaze-medium-diverse and Kitchen-mixed tasks. It is evident that CTP consistently outperforms Diffuser across both AntMaze and Kitchen domains for the same number of denoising steps $N$. This implies that, to achieve a comparable level of performance, CTP can deliver a substantial inference speedup relative to Diffuser, underscoring the scalability advantage of CTP in both navigation and manipulation tasks.

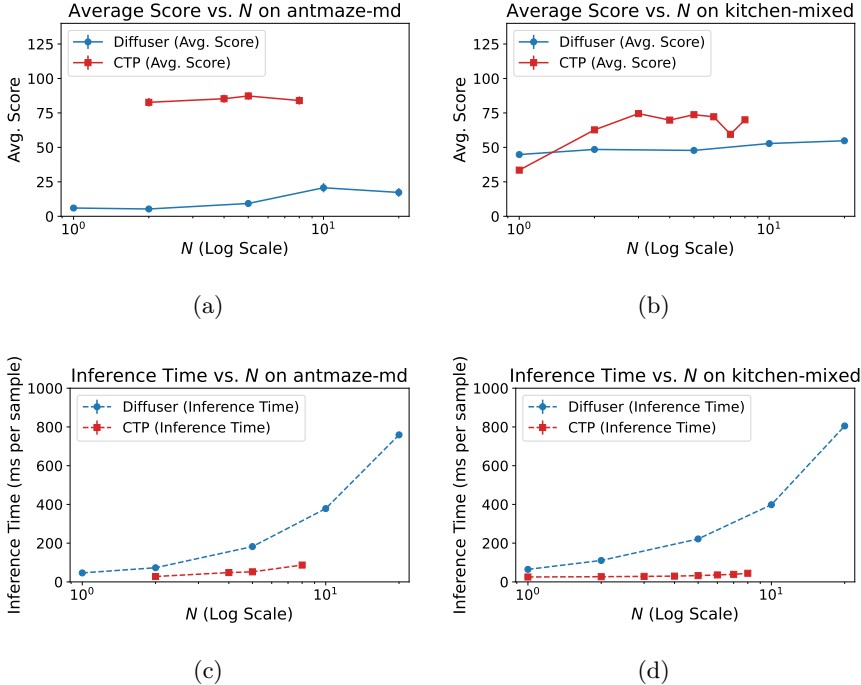

Figure 3: Comparison of average normalized scores (top) and inference times (bottom) of CTP and Diffuser on AntMaze and Kitchen tasks.

These results underscore the efficiency advantage of CTP, demonstrating that it achieves strong performance with substantially fewer denoising steps, thereby offering a more practical solution for time-sensitive deployment scenarios.

**Error accumulation.** Score-based models are prone to discretization errors introduced by SDE/ODE solvers, while distillation-based models often suffer from error accumulation across multiple sampling steps. CTM mitigates these issues and reduce the error accumulation from $O\left(\sqrt{t_1 + t_2 + \cdots + t_N}\right)$ to $O\left(\sqrt{t_N}\right)$ (Kim et al., 2024). As shown in Figure 4, the proposed algorithm CTP achieves significantly improved performance over offline RL methods based on CM when the number of sampling steps $N > 2$.

## 5.4 Ablation Studies

To analyze the contributions of various design components in CTP and isolate the effect of using CTMs, we conduct several ablation and controlled experiments. To isolate the benefits of consistency trajectory modeling, we conduct controlled comparisons against two existing methods: DD (which uses a diffusion

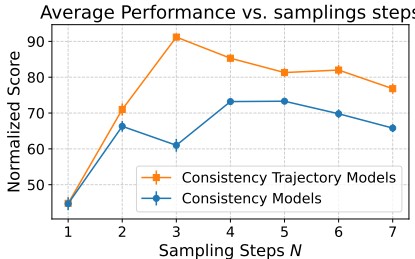

Figure 4: The average performance versus sampling steps $N$ for CTP and CM on kitchen-partial dataset.

model) and CP (which uses a CM). For fairness, we adapt both methods to use the same architecture and training setup as our CTP, including the Transformer-based backbone, stride-based trajectory representation, value-based filtering, and training setup. We refer to these unified-architecture variants as DD-improved and CP-improved respectively. This setup ensures that all methods differ only in the generative models used, allowing for a clean and controlled comparison. The results, shown in Table 6, demonstrate that in identical architectural and algorithmic settings, CTP consistently outperforms the DD-improved and CP-improved in terms of average return in Maze2D tasks under comparable denoising steps. Corresponding results on the Kitchen domain, as well as additional ablation studies on the M-stride trajectory representation and value-based trajectory selection, are provided in Appendix B.

Table 6: Average normalized scores and inference time across different denoising steps $N$ for CTP, CP-improved, and DD-improved in Maze2D environments.

| Method | N | Average Score | | | Inference time (ms) |
|---|---|---|---|---|---|
| | | Maze2D-Large | Maze2D-Medium | Maze2D-Umaze | |
| | 1 | $218.1 \pm 3.5$ | $167.1 \pm 2.4$ | $154.1 \pm 2.3$ | $9.6 \pm 0.1$ |
| CTP | 2 | $212.9 \pm 2.8$ | $165.5 \pm 2.7$ | $149.9 \pm 2.5$ | $11.7 \pm 0.1$ |
| | 3 | $220 \pm 3.5$ | $169.9 \pm 2.4$ | $156.2 \pm 2.2$ | $13.6 \pm 0.1$ |
| | 1 | $185.1 \pm 4.5$ | $150.8 \pm 3.3$ | $147.8 \pm 3.0$ | $10.0 \pm 0.2$ |
| CP-improved | 2 | $204.1 \pm 4.0$ | $153.8 \pm 2.8$ | $145.5 \pm 3.1$ | $11.5 \pm 0.1$ |
| | 3 | $61.2 \pm 7.3$ | $91.8 \pm 7.4$ | $20.5 \pm 6.3$ | $13.2 \pm 0.1$ |
| | 1 | $182.7 \pm 5.4$ | $106.6 \pm 5.4$ | $113.9 \pm 5.1$ | $9.5 \pm 0.2$ |
| DD-improved | 5 | $203.5 \pm 3.5$ | $160.7 \pm 2.7$ | $146.1 \pm 3.1$ | $23.0 \pm 0.2$ |
| | 10 | $207.5 \pm 4.1$ | $159.0 \pm 2.8$ | $142.7 \pm 3.0$ | $40.9 \pm 0.2$ |

## 6 Conclusion

In this work, we propose CTP, a novel approach that integrates CTM into the trajectory optimization framework for offline RL. CTP enables highly effective trajectory sampling with only one or a few denoising steps, achieving superior performance across standard benchmarks. Our method demonstrates robust generalization in both simple and complex environments, showcasing its practical value for efficient policy planning.

Looking ahead, several avenues of improvement remain. First, enhancing the critic model (Kostrikov et al., 2022) to better estimate the upper bound of plausible returns—rather than the mean—can improve planning guidance; this can be achieved using asymmetric loss functions such as the expectile loss. Second, the consistency loss weighting scheme can be refined to more accurately reflect the impact of different noise scales during training (Song & Dhariwal, 2024), thereby amplifying the signal from informative samples. Finally, incorporating advanced network architectures (e.g., UNet, Transformer) along with curriculum learning strategies may further improve training stability and policy performance. We leave these directions for future work.

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

# A   Task Properties

Table 7 provides a comprehensive summary of dataset sizes used in our experiments. All datasets are sourced from the latest release of the D4RL benchmark suite (Fu et al., 2020), encompassing four domains: Gym-MuJoCo, Maze2D, FrankaKitchen, and AntMaze.

In the Gym-MuJoCo domain, we consider three data variants: medium-expert, medium, and medium-replay. The medium-expert datasets comprise trajectories generated by a mixture of medium- and expert-level policies, thus incorporating both suboptimal and near-optimal actions. The medium datasets are collected exclusively using medium-performance policies, and consequently, contain a higher proportion of suboptimal behavior. The medium-replay datasets represent replay buffers collected from medium-level agents during training, featuring highly diverse suboptimal trajectories and exploration noise. The Gym-MuJoCo domain consists of three standard environments:

- **Hopper** involves a single-legged robot that must learn to hop forward without falling, presenting challenges in stability and balance.

- **HalfCheetah** features a planar cheetah-like agent with a more complex morphology, requiring the agent to coordinate multiple joints to achieve high-speed locomotion.

- **Walker2d** simulates a two-legged humanoid robot that must walk forward while maintaining upright posture, making it more prone to instability.

These tasks serve as standard benchmarks for evaluating the ability of offline algorithms to generalize from limited data and produce smooth, efficient motion without direct online interaction.

In the Maze2D domain, datasets are categorized according to maze complexity: umaze, medium, and large. Each variant reflects increasing navigation difficulty and state space dimensionality.

The FrankaKitchen domain includes two types of datasets: partial and mixed, both of which consist of undirected demonstrations. In the partial dataset, a subset of trajectories successfully accomplish the full task, enabling imitation learning algorithms to leverage these informative samples. Conversely, the mixed dataset contains only partial trajectories, with no complete demonstrations of the full task. This requires RL algorithms to generalize from sub-trajectories and effectively compose them into successful task completions.

For the AntMaze domain, we use the same three maze configurations (umaze, medium, and large) as in Maze2D. Three types of datasets are constructed: In the standard setting (antmaze-umaze-v0), the ant is instructed to reach a fixed goal from a fixed start state. In the diverse datasets, both the goal and starting positions are randomly sampled, introducing high variability. In the play datasets, the ant is commanded to reach hand-picked waypoints in the maze, which may not coincide with the evaluation goal, and also begins from a curated set of start locations. These variations are designed to assess the agent's ability to generalize under different levels of distributional shift and task ambiguity.

The Adroit Hand benchmark comprises four distinct manipulation scenarios, each instantiated with a 30-DoF dexterous hand mounted on a freely moving arm:

- **Door.** The agent must disengage a latch with substantial dry friction and swing the door until it contacts the stopper. No explicit latch state is provided; the agent infers its dynamics solely through interaction. The door's initial pose is randomized across trials.

- **Hammer.** The hand picks up a hammer and drives a nail of variable location into a board. The nail—subject to dry friction resisting up to 15N—must be fully embedded for success.

- **Pen.** With the wrist fixed, the agent reorients a blue pen so that its pose matches a randomly placed green target within a prescribed angular tolerance.

- **Relocate.** The agent transports a blue sphere to a green target whose position, along with the sphere's start pose, is uniformly randomized throughout the workspace; success is declared once the sphere lies within an $\epsilon$-ball of the target.

| Domain | Task Name | Samples |
|---|---|---|
| | hopper-me | $2 \times 10^6$ |
| | hopper-m | $10^6$ |
| | hopper-mr | 402000 |
| | halfcheetah-me | $2 \times 10^6$ |
| Gym-MuJoCo | halfcheetah-m | $10^6$ |
| | halfcheetah-mr | 202000 |
| | walker-me | $2 \times 10^6$ |
| | walker-m | $10^6$ |
| | walker-mr | 302000 |
| | maze2d-umaze | $10^6$ |
| Maze2D | maze2d-medium | $2 \times 10^6$ |
| | maze2d-large | $4 \times 10^6$ |
| FrankaKitchen | kitchen-mixed | 136950 |
| | kitchen-partial | 136950 |
| | antmaze-medium-play | $10^6$ |
| | antmaze-medium-diverse | $10^6$ |
| AntMaze | antmaze-large-play | $10^6$ |
| | antmaze-large-diverse | $10^6$ |

Table 7: Size for each dataset is provided. The number of samples indicates the total count of environment transitions recorded in the dataset (Fu et al., 2020).

## B Additional Ablation study

**Effect of applying CTM.** To complement the controlled comparisons reported in the main paper (Table 6), we provide additional results on the Kitchen domain in this appendix. As in the Maze2D experiments, we evaluate three variants—CTP, CP-improved, and DD-improved—under identical architectural and algorithmic settings.

The results, summarized in Table 8, show that CTP consistently outperforms the other two variants in terms of average return on the Kitchen task. Furthermore, CTP achieves comparable or higher performance while requiring fewer denoising steps, demonstrating improved inference efficiency in this more complex, high-dimensional control domain.

Table 8: Average normalized scores and inference time across different denoising steps $N$ for CTP, CP-improved, and DD-improved in Kitchen environments.

| Method | N | Average Score | | Inference time (ms) |
|---|---|---|---|---|
| | | Kitchen-Partial | Kitchen-Mixed | |
| | 1 | $44.8 \pm 0.9$ | $52.7 \pm 2.1$ | $9.7 \pm 0.3$ |
| CTP | 2 | $71 \pm 1.7$ | $64.8 \pm 1.0$ | $11.3 \pm 0.3$ |
| | 3 | $91.2 \pm 1.0$ | $74.5 \pm 0.3$ | $13.3 \pm 0.3$ |
| | 1 | $55.8 \pm 1.9$ | $66.8 \pm 1.1$ | $9.7 \pm 0.3$ |
| CP-improved | 2 | $66.3 \pm 1.1$ | $69 \pm 0.8$ | $11.2 \pm 0.3$ |
| | 3 | $11 \pm 1.0$ | $14.3 \pm 1.0$ | $13.2 \pm 0.4$ |
| | 1 | $0.0 \pm 0.0$ | $23.8 \pm 2.1$ | $9.6 \pm 0.3$ |
| DD-improved | 5 | $54.1 \pm 2.1$ | $52.1 \pm 1.5$ | $24.1 \pm 0.4$ |
| | 10 | $80.1 \pm 1.2$ | $66.5 \pm 1.0$ | $40.0 \pm 0.3$ |

**Effect of the M-stride Trajectory Representation.** We investigate the impact of the stride-based trajectory representation, where only every $M$-th state is sampled from the original sequence. We compare

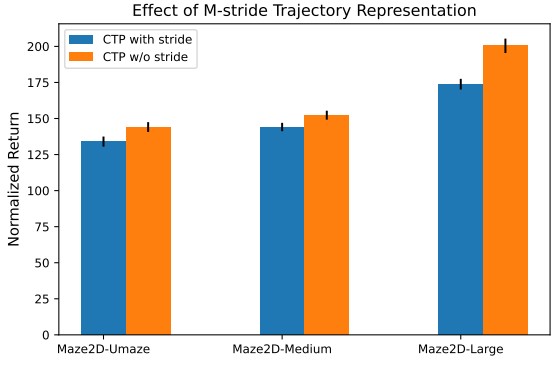
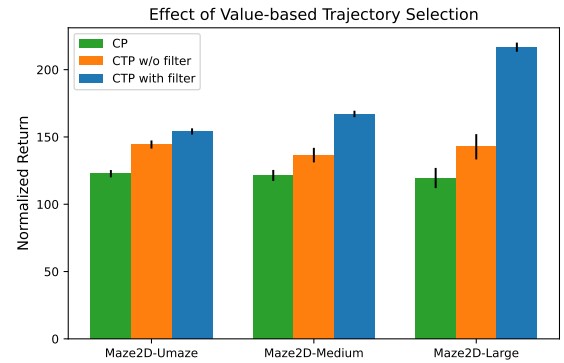

(a) Effect of M-stride trajectory representation.  (b) Effect of value-based trajectory selection.

Figure 5: The ablation study on (a) M-stride trajectory representation and (b) value-based trajectory selection across three Maze2D environments.

our original CTP ($M = 5$, $H = 16$) with CTP (no stride, $H = 80$), both of which correspond to an effective lookahead of 80 steps.

Across all Maze2D tasks, CTP (no stride) achieves the best overall performance, followed by CTP ($M = 5$). These results highlight that introducing the M-stride representation slightly reduces performance but substantially improves computational efficiency, as it reduces the effective planning steps (16 vs. 80). Thus, the stride formulation offers a practical trade-off between accuracy and efficiency.

Quantitative results are shown in Figure 5a.

**Effect of Value-based Trajectory Selection.** Next, we evaluate the influence of the value network–based selection used during inference. We compare CTP with filtering versus CTP without filtering, which uses classifier-free guidance for trajectory generation.

We further include the CP baseline, which also employs classifier-free guidance by default, for comparison.

Results show that CTP with filtering achieves higher returns across all Maze2D variants, followed by CTP without filtering and CP. The results are shown in Figure 5b. This suggests that: (i) under the same classifier-free guidance mechanism, CTP consistently outperforms CP, demonstrating the inherent advantage of CTM in trajectory optimization; and (ii) value-based selection further improves planning quality by aligning trajectory choice with predicted returns, providing a more stable and reward-consistent mechanism than classifier-based guidance.

## C  Implementation Details

This section outlines the architectural and training details used throughout our experiments.

**CTM Training.** This is the teacher–student distillation algorithm for consistency trajectory models (Song et al., 2021; Kim et al., 2024), included here for completeness.

**Model Architecture.** We use Diffusion Transformer blocks with adaLN-Zero architecture (Peebles & Xie, 2023) as the network backbone for the diffusion model and CTM.

**Transformer Depth.** We use Transformer depth of 2 in all the MuJoCo tasks, Kitchen tasks, Maze2D tasks, and Adroit tasks, 8 in AntMaze-Medium tasks, 12 in AntMaze-Large tasks.

**Training Hyperparameters.** The teacher model (EDM) is trained using a learning rate of $2 \times 10^{-4}$, while the student model (CTM) is trained with a smaller learning rate of $8 \times 10^{-6}$. Both models are optimized

---

**Algorithm 2** Consistency Trajectory Model's Training

---

1: **repeat**
2:     Sample $\mathbf{x}_0(\tau) := (s_k, s_{k+M}, \ldots, s_{k+(H-1)M})$ from data distribution
3:     Sample $\boldsymbol{\epsilon} \sim \mathcal{N}(0, I)$
4:     Sample $t \in [\epsilon, t_N], \ w \in [0, t], \ u \in [w, t)$
5:     Calculate $\mathbf{x}_t = \mathbf{x}_0 + t\boldsymbol{\epsilon}$
6:     Calculate $\text{SOLVER}(\mathbf{x}_t, t, u; \boldsymbol{\phi})$
7:     Update $\boldsymbol{\theta} \leftarrow \boldsymbol{\theta} - \frac{\partial}{\partial \boldsymbol{\theta}} \mathcal{L}(\boldsymbol{\theta}, \boldsymbol{\eta})$
8:     Update $\boldsymbol{\eta} \leftarrow \boldsymbol{\eta} + \frac{\partial}{\partial \boldsymbol{\eta}} \mathcal{L}_{\text{GAN}}(\boldsymbol{\theta}, \boldsymbol{\eta})$
9: **until** converged

---

using the Adam optimizer with a batch size of 128 for a total of $1 \times 10^6$ training steps. Additionally, we train the inverse dynamics model and the critic using a learning rate of $3 \times 10^{-4}$, also with the Adam optimizer.

**Inference Settings.** During inference, we set the number of sampling steps $N = 2$ for CTP procedure.

**Planning Horizon and Stride.** We set the planning horizon $H$ and stride $M$ as follows:

- **MuJoCo tasks:** $H = 4$, $M = 1$

- **Kitchen tasks:** $H = 32$, $M = 4$

- **Maze2D tasks:** $H = 32$, $M = 15$

- **AntMaze tasks:** $H = 40$, $M = 25$

- **Adroit tasks:** $H = 32$, $M = 2$

These hyperparameters were selected based on task complexity and temporal resolution, with longer planning horizons and strides for more complex, long-horizon tasks such as AntMaze and Kitchen.

