# OpenReview forum: "Consistency Trajectory Planning: High-Quality and Efficient Trajectory Optimization for Offline Model-Based Reinforcement Learning"
_TMLR — Accepted by TMLR_

### Review · Reviewer_nXd1 · 2025-08-10

**Summary Of Contributions:**

The paper introduces **Consistency Trajectory Planning (CTP)**, an offline model-based RL planner that replaces the iterative denoising process of diffusion planners with a single- or two-step **Consistency Trajectory Model (CTM)** distilled from a score-based teacher. CTP rapidly samples candidate trajectories, ranks them with a learned return critic, and executes the first action via an inverse‑dynamics model.  Experiments on 27 D4RL tasks show comparable or superior returns to diffusion baselines while reducing planning latency by ≈120× on Walker2d.

**Audience:**

Yes

**Audience Explanation:**

**CTP** connects several different research sub-areas in machining learning -- RL, generative model, planning... It is of great interest to different researchers.

**Broader Impact Concerns:**

No direct ethical concerns.  Faster RL planners could accelerate deployment in safety‑critical systems.

**Claims And Evidence:**

Yes

**Claims Explanation:**

1. Bridges recent CTM advances with offline RL planning, addressing the pressing speed bottleneck of diffusion planners.

2.  Evaluated on four domains (MuJoCo, Maze2D, AntMaze, Kitchen) with 150 seeds per task; wall‑clock comparisons included.

3. Distillation loss combines global/local consistency, DSM regularization, and GAN loss to mitigate degradation.

4. One‑shot (or two‑shot) sampling simplifies deployment.

**Requested Changes:**

1. CTP underperforms model‑free Diffusion‑QL on 6/9 locomotion tasks; the cause is not analyzed. So please Analyze the MuJoCo performance gap; consider increasing candidate count *N* or adjusting critic training.

2.  Latency measured only on Walker2d; large maze tasks would better showcase the claimed scalability. Report latency on AntMaze‑Large and Kitchen tasks.

3. Lacks convergence or error‑propagation analysis for CTM sampling.

4. Header artefacts, small fonts in Figure 2. Fix presentation artefacts and enlarge figure fonts.

5. It would be better if the authors could release code and pretrained CTM weights.

---

### Review · Reviewer_eRuK · 2025-09-19

**Summary Of Contributions:**

The paper proposes a model-based offline reinforcement learning method called Consistency Trajectory Planning (CTP) that generates trajectory segments conditioned on the current state for planning. The main contribution of this work is to use a Consistency Trajectory Model (CTM), instead of diffusion models as done in prior work, to reduce the high computational cost of iterative denoising. CTM allows fast denoising using fewer denoising steps while maintaining good sample quality. The proposed method, CTP, involves training a CTM over state trajectory segments, selecting the best plan based on a learned value function, and then using an inverse dynamics model to produce an action given two consecutive states from the generated trajectory. Experiments on D4RL locomotion, Maze2D, AntMaze, Kitchen, and Adroit tasks show that CTP achieves competitive performance compared to existing methods while using much fewer function calls (and hence lower inference time) compared to a diffusion-based method.

**Audience:**

Yes

**Audience Explanation:**

There is quite a lot of interest in applying generative models to RL. This work is a step towards making these methods more viable by reducing the inference compute required, which is a notable limitation of existing methods. I believe the problem addressed in this paper and this particular approach is of interest to a sub-section of the community.

**Broader Impact Concerns:**

This work does have any ethical considerations that need to be addressed, and does not require a Broader Impact Statement.

**Claims And Evidence:**

No

**Claims Explanation:**

1. The main empirical claim seems to be that CTP matches or outperforms existing diffusion-based methods. There are three design choices in CTP that differ from existing methods, particularly diffuser and DD, which seem to be the main point of comparison.
    - Using a CTM instead of a diffusion model to generate trajectories.
    - Generating multiple samples from the model and filtering them based on a learned value function, as opposed to using classifier or classifier-free guidance.
    - Using jump trajectories that produce states every $M$ timesteps, giving it an effective horizon of $H \times M$.

The focus of this work is clearly on incorporating CTM into the pipeline, but there are no experiments that control for the other two factors. The other two aspects, i.e., filtering multiple samples and generating every $M$th state, can be easily applied to diffusion-based methods as well. Without such experiments, it is not possible to determine whether using CTM alone is enough to match the performance of diffusion-based methods without the other modifications. It has been shown in previous work [2] that generating multiple samples and filtering seems to generally be better (this relevant paper should also be cited), and I suspect having a longer horizon of $M \times H$ could be an important factor in tasks like Maze2D and AntMaze.

2. The entirety of Sections 2 and 3 is very similar to the discussion in [1], apart from some light rephrasing and rewording. The authors should also clearly articulate what separates their method from [1] since there are quite a few similarities.

3. There are some parts of the introduction that do not have supporting citations or do not seem to apply to CTP.
    - “As a result, in practice, the multistep sampling procedure of CM … accumulates errors particularly as the number of function evaluations increases.” This statement is provided as is without supporting evidence or citations. This is important because using CTM instead of CM seems to be the key difference from [1].
    - The introduction and several places in the paper mention “single-step sampling” as one strength of CTP, but in practice, the authors use different number of denoising steps depending on task complexity. Therefore, this phrase should be changed throughout the paper.

*[1] Wang, Guanquan, Takuya Hiraoka, and Yoshimasa Tsuruoka. "Planning with consistency models for model-based offline reinforcement learning." Transactions on Machine Learning Research (2024).*

*[2] Hansen-Estruch, P., Kostrikov, I., Janner, M., Kuba, J.G. and Levine, S., 2023. Idql: Implicit q-learning as an actor-critic method with diffusion policies. arXiv preprint arXiv:2304.10573.*

**Requested Changes:**

1. Please provide ablation studies on CTP to control for factors other than using a CTM instead of a diffusion model. Alternatively, the value network-based selection of trajectories and sampling every $M$th state in the sequence can be applied to all of the diffusion trajectory methods, including diffuser, DD, and CP.

2. Please rewrite Section 2 and 3 substantially to include original content. Also, currently the background section discusses CM but not CTM, which is puzzling the latter is relevant to this work but the former is not. Without any background discussion on CTM, it is confusing to read about the method in Section 4 as teacher and student networks are mentioned without any context.

3. Most of Section 4 is related to general CTM training, and is not specific to CTP. I suggest the authors move the discussion on CTM to the background section, and reserve Section 4 only for CTP-specific discussion. Also, currently the description of CTM and its losses is too verbose, it can be made much shorter since it is part of existing work.

4. Please clearly differentiate CTP from the closely related existing work CP [1]. It is currently included as a baseline, but there is no discussion of this work in the related section or the experiment section.

5. The related work section can be expanded to include many more existing works, including seminal work, on diffusion models applied to RL. There is a very rich literature in this area and the paper does not provide enough context to an uninformed reader. A few important works that may be included are [2,3,4,5,6].

*[1] Wang, Guanquan, Takuya Hiraoka, and Yoshimasa Tsuruoka. "Planning with consistency models for model-based offline reinforcement learning." Transactions on Machine Learning Research (2024).*

*[2] Chi, Cheng, Zhenjia Xu, Siyuan Feng, Eric Cousineau, Yilun Du, Benjamin Burchfiel, Russ Tedrake, and Shuran Song. "Diffusion policy: Visuomotor policy learning via action
diffusion." The International Journal of Robotics Research (2023): 02783649241273668.*

*[3] Hansen-Estruch, P., Kostrikov, I., Janner, M., Kuba, J.G. and Levine, S., 2023. Idql: Implicit q-learning as an actor-critic method with diffusion policies. arXiv preprint arXiv:2304.10573.*

*[4] Psenka, Michael, Alejandro Escontrela, Pieter Abbeel, and Yi Ma. "Learning a diffusion model policy from rewards via Q-score matching." In Proceedings of the 41st International Conference on Machine Learning, pp. 41163-41182. 2024.*

*[5] Jain, Vineet, Tara Akhound-Sadegh, and Siamak Ravanbakhsh. "Sampling from Energy-based Policies using Diffusion." In Reinforcement Learning Conference, 2025.*

*[6] Wang, Yinuo, Likun Wang, Yuxuan Jiang, Wenjun Zou, Tong Liu, Xujie Song, Wenxuan Wang et al. "Diffusion actor-critic with entropy regulator." Advances in Neural Information Processing Systems 37 (2024): 54183-54204.*

---

> ### Comment · Reviewer_eRuK · 2025-11-09
> **Clarifying new ablations and results**
>
> Hi authors,
>
> Thanks for the modifications, particularly rewriting Sections 2-4. The ablation studies are appreciated, but it misses the point so let me rephrase my request in a different way:
> - The main question in my review was "How does CTP perform compared to other methods, if we keep everything same apart from the use of CTMs?". This means using the same horizon length, strides, conditioning mechanism, and architectures.
> - The comparison of H=16, stride=5 versus H=80, no stride is insightful. But it still does not answer the question - how does CTP compare with diffuser/decision diffuser if both have the same horizon/stride settings?
> - The ablation of filtering versus conditioning shows that conditioning results in a substantial drop in performance, making it comparable to existing methods in Table 2. What if diffuser/CP/RACTD also used filtering based on Q-values?
> - CTP uses a diffusion transformer architecture, whereas diffuser/decision diffuser use a temporal U-net. This was a detail that I missed in my review, could the authors control for this variation in architecture?
>
> I guess my main concern across all the above points is **fairness of comparison with baselines**. CTP uses different architectures and hyperparameters/algorithmic changes that could be applied to baselines as well. So it is difficult to judge how much well CTM performs by itself. Can the authors compare with baselines while making sure the model architecture and other settings are kept the same apart from replacing diffusion models with CTMs?

---

> > ### Author Response · Authors · 2025-11-17
> > **Response to Reviewer’s Comment**
> >
> > We thank the reviewer for the clear and constructive clarification. We now better understand that the core concern is whether the performance gain of CTP stems from the use of CTMs alone, or from other design choices such as architecture, stride, or value-based filtering.
> >
> > In response, we have added a new controlled ablation experiment to directly address this point. The results are reported in Table 6 (Section 5.4) for the Maze2D tasks, and in Table 8 (Appendix B) for the Kitchen domain, which is included in the appendix due to space limitations.
> >
> > Specifically, we compare CTM with a diffusion model and a consistency model under identical settings, keeping the following components fixed: transformer-based architecture, horizon length and stride, value-based filtering, training procedure and dataset. The only difference lies in the choice of generative model: CTM vs. diffusion vs. consistency model. The results show that CTM consistently outperforms the diffusion and consistency model baseline across Maze2D and Kitchen tasks under comparable denoising steps or comparable inference time.
> >
> > We hope this addresses the reviewer’s concern about the fairness of comparison and strengthens the empirical support for the CTM framework as the primary driver of improvement.

---

### Review · Reviewer_oZMQ · 2025-11-05

**Summary Of Contributions:**

This paper introduces Consistency Trajectory Planning (CTP), a model-based offline RL algorithm that leverages Consistency Trajectory Models (CTMs) for trajectory optimization. The authors report strong performance and efficiency on diverse benchmarks, especially in long-horizon, goal-conditioned tasks. However, the method appears to be a relatively direct application of CTMs to prior work, and its advantages over existing approaches are not clearly shown.

**Audience:**

Yes

**Audience Explanation:**

The application of consistency models to RL planning is an interesting direction, and RL planning remains a popular topic in machine learning.

**Claims And Evidence:**

No

**Claims Explanation:**

- Claim "CTP is highly efficient, achieving over 120× speedup in inference time."

The baseline used, Diffuser, is known for its high computational cost and not a state-of-the-art; a comparison against state-of-the-art methods is needed for a meaningful assessment of efficiency.

- Claim "CTP consistently outperforms existing diffusion-based planning methods"

Table 1 shows D-QL has better performance and D-QL is notably absent from comparisons in Tables 2, 4, and 5. An explanation for its omission would help to fully support the claim of consistent outperformance.

**Requested Changes:**

1. The inference speed comparison should be extended to include more state-of-the-art (SOTA) algorithms, especially methods that are claimed to be more computationally efficient than Diffuser.
2. Add results for D-QL (and maybe also other omitted methods) into all comparative tables, or provide a clear justification for their exclusion to ensure consistent evaluation.

---

> ### Author Response · Authors · 2025-11-17
> **Response to Reviewer’s Comment**
>
> We thank the reviewer for the detailed and constructive feedback. We have carefully addressed all the concerns as follows:
>
> 1.	Concern regarding the inference speed comparison
>
> We appreciate the reviewer for raising this important concern regarding the efficiency comparison. We acknowledge that Diffuser is computationally expensive, and that comparing solely against it may not fully support the claim of high efficiency. In response, we have revised our main claim in abstract and Section 5.3.
>
> To provide a more meaningful evaluation, we have conducted a controlled comparison that includes adapted versions of both Decision Diffuser and Consistency Planning (Section 5.4), which are known to be more efficient diffusion- and consistency-based planners. We refer to these variants as \textbf{DD-improved} and \textbf{CP-improved}, respectively. These variants are implemented under the same settings as CTP: Transformer-based architecture, identical horizon length and stride, value-based filtering, and training setup.
>
> The corresponding inference times and average scores are presented in Table 6 (Section 5.4) and Table 8 (Appendix B, due to space limitations). These results show that when achieving comparable performance, CTP requires fewer denoising steps and less inference time.
>
> We called them DD-improved and CP-improved because we use a faster and improved version of them by keeping the following components fixed: transformer-based architecture, horizon length and stride and value-based filtering. The inference time and corresponding average score are presented in Table 6 (Section 5.4) and Table 8 (Appendix B due to space limitations). The results indicate that CTP achieves better performance than DD-improved and CP-improved under comparable inference-time settings, highlighting its efficiency advantages relative to existing diffusion- and consistency-based planners.
>
> 2.	Add results for D-QL in all comparative table.
>
> We appreciate the reviewer’s emphasis on completeness of in baseline comparisons. In the original submission, D-QL was omitted from Tables 2, 4, and 5. In this revision, we have included D-QL results in all three tables. Our updated results show that CTP is competitive with or outperforms D-QL in most tasks, particularly in settings with long-horizon.

---

### Decision · Action_Editor_2sgM · 2025-12-19

**Recommendation:** Accept as is

**Audience:**

Yes

**Audience Explanation:**

The work presents a small but significant improvement on a problem of broad interest.

**Claims And Evidence:**

Yes

**Claims Explanation:**

The paper presents convincing evidence that their approach yields an improvement in terms of inference efficiency w.r.t. existing diffusion-based planning methods.
The contribution of the several components of the proposed approach were initially unclear, but this was already addressed in the revised manuscript.